# Coordination of gaze and action during high-speed steering and obstacle avoidance

**Nathaniel V. Powell[1,2], Xavier Marshall[1], Gabriel J. Diaz[3], Brett R. Fajen[1]\***

**1** Cognitive Science Department, Rensselaer Polytechnic Institute, Troy, New York, United States of America, **2** Center for Perceptual Systems, University of Texas at Austin, Austin, Texas, United States of America, **3** Center for Imaging Sciences, Rochester Institute of Technology, Rochester, New York, United States of America

\* fajenb@rpi.edu

**Data Availability Statement:** All data files are available on Open Science Framework (https://osf.io/5dtxh/).

**Funding:** This material is based upon work supported by the National Science Foundation

## Abstract

When humans navigate through complex environments, they coordinate gaze and steering to sample the visual information needed to guide movement. Gaze and steering behavior have been extensively studied in the context of automobile driving along a winding road, leading to accounts of movement along well-defined paths over flat, obstacle-free surfaces. However, humans are also capable of visually guiding self-motion in environments that are cluttered with obstacles and lack an explicit path. An extreme example of such behavior occurs during first-person view drone racing, in which pilots maneuver at high speeds through a dense forest. In this study, we explored the gaze and steering behavior of skilled drone pilots. Subjects guided a simulated quadcopter along a racecourse embedded within a custom-designed forest-like virtual environment. The environment was viewed through a head-mounted display equipped with an eye tracker to record gaze behavior. In two experiments, subjects performed the task in multiple conditions that varied in terms of the presence of obstacles (trees), waypoints (hoops to fly through), and a path to follow. Subjects often looked in the general direction of things that they wanted to steer toward, but gaze fell on nearby objects and surfaces more often than on the actual path or hoops. Nevertheless, subjects were able to perform the task successfully, steering at high speeds while remaining on the path, passing through hoops, and avoiding collisions. In conditions that contained hoops, subjects adapted how they approached the most immediate hoop in anticipation of the position of the subsequent hoop. Taken together, these findings challenge existing models of steering that assume that steering is tightly coupled to where actors look. We consider the study's broader implications as well as limitations, including the focus on a small sample of highly skilled subjects and inherent noise in measurement of gaze direction.

## Introduction

In the long history of research on the topic of steering through cluttered environments, scientists have studied visual control in the context of a variety of tasks, such as walking through crowded spaces, walking over complex terrain, and driving a car. In general, the focus has been on visual-locomotor tasks that humans do often and with relative ease, and for good

(https://www.nsf.gov/) under Grant No. 2218220 to BRF and by the Office of Naval Research (https://www.nre.navy.mil/) under Grant No. N00014-18-1-2283 to BRF. Any opinions, findings, and conclusions or recommendations expressed in this material are those of the authors and do not necessarily reflect the views of the National Science Foundation or the Office of Naval Research. The funders had no role in study design, data collection and analysis, decision to publish, or preparation of the manuscript.

**Competing interests:** The authors have declared that no competing interests exist.

reason. However, new insights can also be gleaned by studying what humans are capable of doing when pushed to the extreme.

One example of a visual-locomotor task that pushes the limits of human performance is first-person view (FPV) drone racing. This task involves using a remote controller to steer a quadcopter through a densely cluttered environment based on a video feed from a drone-mounted camera to a head-mounted display that is worn by the pilot. The FPV drone piloting task provides a novel but rich context within which to study high-speed visual control of locomotion by skilled actors and is ripe for exploration of numerous aspects of visual control. In the present study, the focus was on gaze and control strategies for steering through waypoints, following a path, and avoiding obstacles.

## Gaze and steering in automobile driving

A key component to the skillful negotiation of complex environments in humans is the ability to actively direct one's gaze to reveal task-relevant information and to coordinate gaze behavior with movement [1–4]. The critical role of active gaze is well established for many perceptual-motor skills, including sports such as cricket, racquetball, and baseball [5–7], walking over complex terrain [8, 9], and steering an automobile along a winding road [10–12].

In studies of gaze behavior during automobile driving, a common finding is that drivers spend much of their time looking at the road and road edges. This has led to the formulation of various hypotheses about why drivers look where they do. The most well-known is the tangent point hypothesis from [10], which states that drivers fixate the tangent point of the upcoming bend because the visual direction of that point specifies the upcoming road curvature [13, 14]. More recent evidence favors an alternative hypothesis, which is that drivers fixate waypoints along their desired future path. During waypoint fixation, information from both the visual direction of the fixated waypoint and the surrounding retinal flow specifies whether one is on track to steer through the waypoint [15, 16]. As such, drivers could simply fixate points on the road through which they want to travel, essentially using the gaze control system to select waypoints [4, 16]. The information that is made available during such fixations can be used to adjust steering, such that the resulting trajectory passes through the waypoints.

The picture of steering that is painted by this hypothesis entails a close coupling of gaze and steering. It is one in which gaze is directed precisely toward locations where drivers want to steer, and steering is biased toward where drivers look. Evidence of the former was reported by [16], who found that subjects looked at different parts of the road depending on where in the lane (inside, center, outside) they were instructed to drive. Likewise, [17] provided evidence that people steer in the direction that they look by instructing subjects to look at a fixation point at one of several lateral positions on the road. On wider roads, trajectories were biased toward the inside edge when the fixation point was positioned toward the inside of the curve and vice-versa.

Taken together, these studies support a theory of steering that is aptly summarized by the phrase "look where you want to go" [4, 16]. From this perspective, the availability of information and hence the successful control of steering depend critically on the precise fixation of certain key reference points on the upcoming road or road edges. This presupposes that the upcoming road is plainly visible and easy to find even as the actor is continually moving through the scene. This condition is met in almost all previous studies of gaze behavior during steering as oftentimes, the only features that are present in the environment are road edges on a flat open planar ground surface, allowing for the upcoming road to be easily discriminated from the background. However, humans and other animals also move through densely cluttered environments, where the presence of other objects may occlude the upcoming path,

there may be shadows that obscure the path, and the appearance of the path may be similar to that of other parts of the scene. In such scenarios, fixations may often fall off the desired future path because actors may need to visually scan the scene to find the path within the clutter.

This raises several questions about the relation between gaze and steering. First, if the information that is used to control steering is normally made available by fixating points on the desired future path, how is the ability to control steering affected when consistent fixation of the future path is not possible because the future path does not immediately pop out? Second, how good are people at finding the upcoming path in scenes that contain dense clutter? Third, dense clutter may not only visually occlude the upcoming path but also serve as an obstacle that has to be circumvented. If precise fixation of points on the future path is critical for following a path, is precise fixation of obstacles also critical for successful obstacle avoidance? Lastly, navigation through densely cluttered environments sometimes involves steering through a series of openings rather than following an explicit path. In such situations, the goal (i.e., the opening) is not an actual object to which gaze can be anchored and therefore fixation of a point on the future path is not possible. Where do people look under such conditions?

## Look-ahead fixations and anticipatory steering adjustments

The need to search for navigable spaces within the clutter may not be the only reason why gaze is not consistently locked onto the future path. Closely related to this is the idea that in certain situations, people might look ahead and anticipate the trajectory adjustments needed to reach goals that lie beyond the immediate future [18, 19]. The ability to change speed and direction to avoid collisions or steer through waypoints is limited by factors such as sensorimotor delays, neuromuscular lags, limitations on action capabilities, and inertial constraints. If actors waited until they passed the most immediate waypoint before taking future waypoints into consideration, they may be forced to make large or erratic trajectory adjustments, may miss an upcoming waypoint, and they may collide with obstacles. As such, performing such tasks well would seem to require actors to look ahead and to adapt their trajectory in anticipation of having to reach successive waypoints.

One of the few studies on this topic was conducted by [19]. They had subjects steer along a course through a series of gates in a virtual environment, similar to slalom skiing. They found that subjects tended to fixate and track the nearest gate until it was within a certain range (about 1.5 s), and then make a look-ahead fixation to the next gate. Occasionally, subjects looked back to the nearest gate before once again shifting gaze ahead to the subsequent gate. They referred to such behavior as *gaze polling* and proposed that it plays a critical role in setting up a smooth trajectory through successive gates. However, analyses of steering behavior suggested that only some subjects followed trajectories that took future gates into consideration while others appeared to focus only on the most immediate gate. Furthermore, for subjects who did adapt their trajectory to anticipate future waypoints, steering behavior was only slightly smoother. Lastly, manipulations that affected how far in advance upcoming gates appeared had at most weak effects on steering smoothness.

Thus, whereas the execution of look-ahead fixations was a robust finding in [19] and has been reported in other studies [e.g., 18], the evidence for anticipatory steering adjustments is less clear. The second aim of this study was to determine whether subjects made anticipatory steering adjustments.

## The FPV drone piloting task

Rather than focusing on the experimental task of automobile driving as others have done for decades, we chose the task of first-person view (FPV) drone racing. This task involves steering

a quadcopter through a densely cluttered environment based on a video feed from a drone-mounted camera to a head-mounted display that is worn by the pilot. The pilot uses a remote controller to adjust the thrust, altitude, yaw rate, and roll of the quadcopter.

The drone piloting task is ideal for the purposes of the present study because it provides a naturalistic context in which humans move at high speeds through densely cluttered environments without risk of injury. In addition, the likelihood of observing anticipatory steering adjustments may be greater because the motion of drones is significantly constrained by inertia, making rapid trajectory adjustments more difficult, and because subjects were skilled drone pilots with extensive experience maneuvering drones in tight spaces.

While the drone piloting task is in many ways an ideal context for studying high-speed steering and obstacle avoidance, studying drone piloting in the real world poses some significant challenges: tracking the movement of the drone in a large space, limited ability to control and manipulate the environment, safety of both human subjects and equipment, dependence on weather (wind, rain), varying amount of sunlight. As such, we developed a drone-piloting simulator. The virtual environment was created using the Unity game engine and was visually realistic and highly customizable. Subjects viewed the scene through an HTC Vive Pro head-mounted display (HMD), which was equipped with a Pupil Labs eye tracker that enables eye tracking at up to 200 Hz. Steering and speed are controlled using a standard RC controller of the kind that is typically used by FPV drone pilots. The flight dynamics were not identical to a real-world racing drone, but they closely approximated a fast-moving photography drone, allowing experienced drone pilots to skillfully maneuver the drone after just a few minutes of practice.

To the best of our knowledge, the only previous study of gaze behavior in drone pilots is [20]. They had subjects repeatedly steer through a series of gates positioned along a figure-8 track. Subjects consistently directed gaze toward the upcoming gate and shifted gaze toward the next gate as soon as they passed through the previous one. They also found strong cross-correlations between gaze angle, camera angle, and thrust angle. The temporal offsets at peak correlation suggested that gaze angle changes preceded camera angle changes, which preceded thrust angle changes. These findings further reinforce the coupling of gaze and steering and provide additional evidence of look-ahead fixations, but the study was not designed to investigate questions about gaze behavior during navigation through cluttered environments.

## The present study

We sought to study gaze and steering behavior under conditions that closely resemble the naturalistic task that drone pilots perform. As such, rather than conducting a traditional experiment with hundreds of individual trials each of which lasted a few seconds, we had subjects steer continuously along a racecourse. The course was an irregularly shaped closed loop embedded in a forest-like virtual environment and took approximately 60–75 s to complete. Each experimental session comprised five blocks, each of which consisted of two laps per condition. Subjects traveled in a clockwise direction in Blocks 1–4 and a counterclockwise direction in Block 5.

Experiment 1 included three conditions. The Path Only condition was most similar to automobile driving but with nearby trees that served as obstacles and occluded the path (Fig 1A). In the Hoops Only condition, hoops were placed at various locations along the path but the path was not rendered (Fig 1B). The task was to steer through the hoops. In the Hoops and Path condition, both features were present, and subjects were instructed to follow the path and steer through hoops while avoiding collisions with nearby trees. In Experiment 2, the conditions were similar to those in the Path Only condition of Experiment 1, but we explicitly

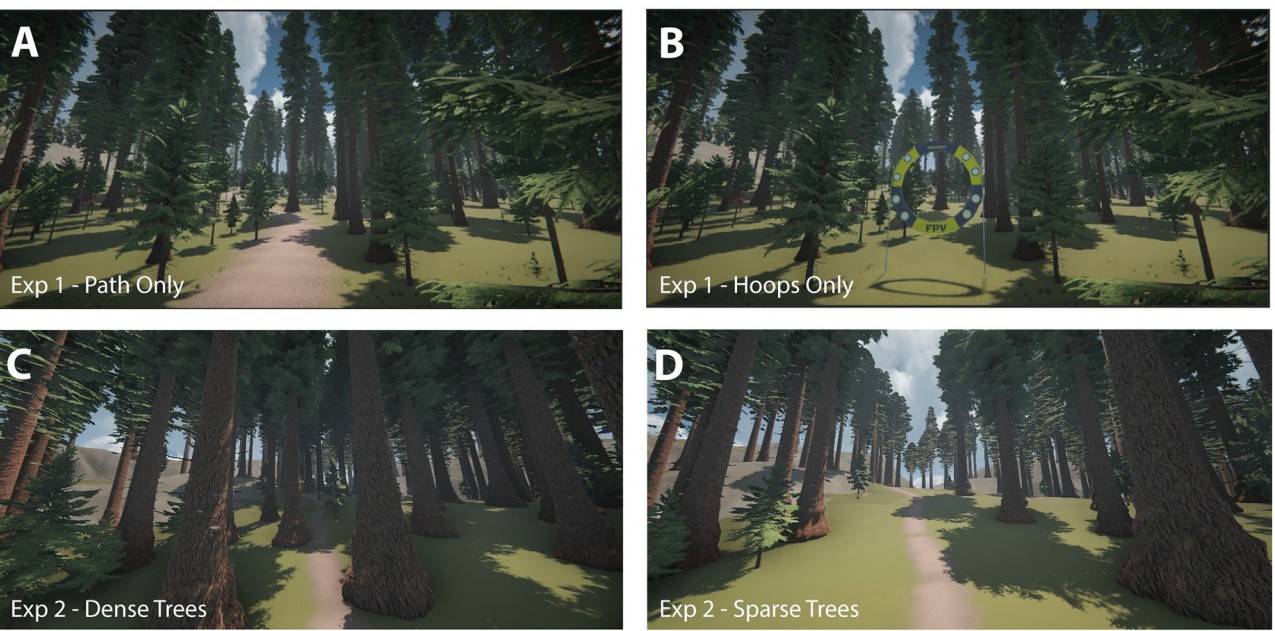

**Fig 1. Screenshots of the Path Only (A) and Hoops Only (B) conditions from Experiment 1 and the Dense Trees (C) and Sparse Trees (D) conditions from Experiment 2.** Experiment 1 also included a Hoops and Path condition (not pictured) that combined the two key features of the Path Only and Hoops Only condition.

manipulated the density of trees on and near the path. Fig 1C and 1D show screenshots from the two conditions (Dense Trees and Sparse Trees, respectively) in that experiment.

## Methods and materials

### Participants

Sample size was limited because a very small percentage of the population has FPV drone piloting experience. In addition, Covid-related precautions prohibited recruitment of subjects from outside the RPI community. Six subjects (ages 19–29 years) participated in Experiment 1 and three subjects (all of whom had completed Experiment 1) participated in Experiment 2. Subject recruitment took place between September 9, 2021 and February 7, 2022. One of the three subjects who participated in both experiments was a co-author of the study. All subjects were experienced quadcopter pilots with at least 10 hours (mean = 108 hours, range = 10–250 hours) of flight time in FPV drone piloting, either real or simulated. All subjects had normal or corrected-to-normal vision and no known visual or motor impairments.

### Hardware

The experiments were conducted using an Alienware Aurora R12 desktop PC running Micro-soft Windows 10 and equipped with an Intel i9 (11 series) ten-core processor, NVIDIA GeForce RTX 3090 24GB graphics processor, and 32GB of 3400 MHz DDR4 RAM. The environment was viewed through an HTC Vive Pro head-mounted display (HMD) with head tracking turned off to mimic the conditions experienced by FPV drone pilots. The HTC Vive Pro has a total resolution of 2160x1200 (1080x1200 per eye), a refresh rate of 90Hz, and 110-degree horizontal and vertical fields of view. A Pupil-Labs VR/AR extension eye tracker [21] was mounted inside of the HMD. The eye tracker was set up so that each eye recorded at

120Hz with a resolution of 196x196. The simulated quadcopter was controlled using a Taranis Q X7 RC controller with a micro-USB connection.

## Virtual environment

The virtual environment was designed using the Unity game engine (version 2019.3.14f) and minimally consisted of a ground surface, trees, and a partly cloudy sky. The ground surface was created using a terrain generation tool (Gaia Pro, which was purchased from the Unity Asset Store) and included natural-looking ground textures and small variations in elevation. The terrain was discretized into a grid-like pattern and coniferous trees of various types and sizes were placed at random positions within each section of the grid. If trees were spawned too close to the path or too close to one another, they were moved by hand in the Unity editor.

This base environment was augmented to create different experimental conditions by changing aspects of the scene, such adding a path, placing hoops along the path, and changing the locations and density of trees. The path was present in the Path Only and Hoops + Path conditions of Experiment 1 as well as in both conditions of Experiment 2. The path was hand-crafted using the Path Painter II Asset to form an irregularly shaped loop with a width of ~1 m and a circumference of approximately 740 m. In the Hoops Only and Hoops + Path conditions of Experiment 1, 42 yellow hoops (~1 m in diameter) were placed at irregular intervals along the path. The Unity Experiment Framework (UXF) [22], an open-source Unity-based experimental interface tool was used to move subjects through each phase of the experiment.

The quadcopter's physics relied on two assets from the Unity store: the "red" drone from the Drones Bundle Package for the physical object and FPV Drone Controller for the controls. The quadcopter mirrored a typical photography drone with the maximum speed set to approximately 17 m/s. The left joystick controlled the altitude (Y axis) and the yaw rate (X axis) while the right joystick controlled forward/backward thrust (Y axis) and lateral (X axis) thrust. The quadcopter model automatically compensated for the force of gravity such that it stabilized at a fixed altitude as long as the left joystick was centered along the Y axis. If the vehicle collided with a tree, hoop, or the ground, it bounced back for a brief period, after which time normal flight resumed. Rarely, the quadcopter would get stuck in the branches of a tree. When this happened, the experimenter aborted the current lap and restarted the drone at the beginning of the course.

## Design and procedure

In both experiments, subjects first signed written consent after which they listened to a set of brief instructions. They were then asked to place the HMD on their head and adjust the head straps so that they could see the full view of the screen in the HMD and so the eye tracker camera had adequate coverage of their eyes. Subjects then followed the eye tracker calibration process as described in the subsection on eye tracking below. Next, subjects completed a practice phase during which they flew freely in an open environment to familiarize themselves with the controls and learn how the virtual quadcopter responded to joystick inputs. After three minutes, they were asked if they were sufficiently comfortable with the controls to continue on to the testing blocks. If they were not yet comfortable, they were given additional time to continue practicing until they were ready for testing.

The main part of Experiment 1 comprised five blocks, each of which included three sub-blocks. Within each sub-block, subjects completed two laps around the loop in each of the three conditions (Hoops Only, Hoops + Path, Path Only), yielding 10 laps per condition across the entire experiment. The Hoops Only and Path Only condition alternated between the first and third sub-blocks and the Hoops + Path condition was always the second sub-block.

Subjects were asked to recalibrate the eye-tracker at the beginning of each sub-block. The quality of each calibration was assessed and recorded.

Subjects were instructed to fly as quickly as they could while staying close to the path (in the Path Only and Hoops + Path conditions), attempting to pass through each hoop (in the Hoops Only and Hoops + Path conditions), and avoiding collisions. There was nothing in the instructions or the task itself that indicated to subjects that they should prioritize speed over accuracy or vice-versa. They were further instructed that if they missed a hoop, they should continue on rather than backtrack. If the quadcopter collided with any object in the scene (e.g., tree, branch, leaves, the ground), its velocity was altered to simulate bouncing off the object. In the case of a head-on collision, the quadcopter bounced backwards and its velocity sharply dropped for a brief period. If the quadcopter collided with a tree and became stuck, the experimenter pressed a button to abort the trial and restart from the beginning of the current lap. This only occurred twice in Experiment 1 and never in Experiment 2. The simulation was designed such that if the quadcopter strayed too far off the path (approximately 10 meters to the left or right of the path and 20 meters in the vertical direction), it was automatically teleported to the start of their current lap and the recorded data for that lap was overwritten. However, this never occurred in any sessions of either experiment. In Blocks 1–4, subjects traveled around the loop while moving in the clockwise direction. In the fifth block, subjects started each condition facing the reverse direction and were instructed to fly in the opposite (counterclockwise) direction.

Experiment 2 was identical to Experiment 1 with the exception that there were two conditions (Dense Trees and Sparse Trees) rather than three. Both conditions were similar to those in the Path Only condition of Experiment 1, but the density of trees on and near the path was manipulated. As such, each of the five blocks included two sub-blocks, one for the Sparse Trees condition and one for the Dense Trees condition. Within each block, subjects completed two laps per condition for a total of ten laps per condition across the entire experiment. The order of conditions was reversed in consecutive blocks to minimize order effects. All other aspects of the procedure were the same as in Experiment 1.

## Simulator sickness

It is well established that certain individuals experience symptoms of simulator sickness in virtual environments, especially during simulated self-motion [23, 24]. In the present study, the risk of simulator sickness was minimized because the subjects had previous experience with FPV drone piloting. During pilot testing, we did find that a small number of subjects with fewer hours of previous FPV drone piloting experience did encounter mild symptoms of simulator sickness within the first 5–10 minutes of the experiment. In these cases, the experiment was stopped immediately.

## Eye tracking

Eye movements were recorded and post-processed using the Pupil-Labs Core software. Eye tracker calibration was performed using an 18-point depth-mapped calibration routine provided by Pupil-Lab open-source software. In addition to calibration, an assessment routine was created that used nine points that cover the visual field to determine the accuracy and quality of calibration. After calibration assessment, the average eye-tracking error in visual degrees was displayed on the screen. The experimenter then chose to move the participant onto the experiment or have them recalibrate. If recalibration was chosen, the previous calibration recording was overwritten. The mean eye tracking error exceeded five degrees for three of the six subjects in Experiment 1. The data from these subjects were excluded in the analyses of

gaze behavior (but not from the analyses of performance measures or anticipatory steering behavior). For the remaining three subjects in Experiment 1, the mean eye tracking error was 3.05 degrees in Experiment 1. Error was greatest in the top left and right sectors and the lowest in the center of the display (mean = 2.55 degrees), which is where subjects tended to spend the most time looking. For the three subjects in Experiment 2, the mean eye tracking error was 3.80 degrees (mean = 2.19 degrees in the center). Two of the three subjects in Experiment 2 were among the three from Experiment 1 with accurate gaze data.

## Data analyses

Analyses were based on raw data from the following sources: (1) the Pupil Labs eye tracker (recorded at 120 Hz), (2) the RC controller joystick positions (90 Hz), (3) the drone position and orientation (90 Hz), and (4) the positions and orientations of hoops.

To prepare the data for analysis, the raw data were used to compute the values of several variables. The direction of the thrust vector in the horizontal plane was calculated based on the offset of controller's right joystick on both the X and Y axes (which determined forward/backward and lateral motion). The instantaneous heading direction was calculated based on the change in drone position over consecutive frames. The direction of the forward-facing camera was derived directly from the drone's yaw angle. The approach angle of the quadcopter relative to the upcoming hoop was calculated based on the angle between the instantaneous heading vector and the normal vector passing through that hoop.

In the Hoops Only and Hoops and Path conditions, flight trajectories were parsed into segments that were defined by pairs of consecutive hoops. The first frame in each segment corresponded to the first timestep after the drone passed through the previous hoop and the last frame corresponded to the time step before it passed through the next hoop.

All repeated-measures ANOVAs were run using ezANOVA from the ez package in the R programming language. If the sphericity assumption was violated (as indicated by a significant Mauchly's test), the Greenhouse-Geisser correction was applied to the degrees of freedom. In all figures, error bars correspond to 95% confidence intervals around means with between subject variation removed. We do not report statistical tests for the measures of gaze behavior due to the fact that we only obtained accurate gaze data from three of the six subjects. All linear mixed effects models used to conduct the analysis of anticipatory steering were run using the lmer function from the LME4 package in the R programming language.

## Results

### Basic performance measures

Before analyzing gaze behavior, we will examine four measures of task performance to determine how successfully subjects were able to perform the task in each condition and how their performance changed over blocks. Subjects were instructed to fly as quickly as possible, so speed of travel is one of the key metrics of task performance. The mean speed of travel averaged over blocks and conditions ranged from 8.37 m/s for the slowest subject to 15.00 m/s for the fastest subject. The overall mean speed was 12.19 m/s, which is equivalent to 46.4 km/h or 28.9 mph.

A two-way repeated-measures ANOVA revealed significant main effects of condition ($F_{1.02, 5.12}$ = 7.57, p = .039, $\eta^2$ = 0.60 [0.16, 1.00]) and block ($F_{1.33, 6.64}$ = 10.75, p < .05, $\eta^2$ = 0.68 [0.42, 1.00]) as well as the condition x block interaction ($F_{3.01, 15.06}$ = 6.45, p < .01, $\eta^2$ = 0.56 [0.32, 1.00]) (see Fig 2A). Bonferroni corrected contrasts revealed that speed was slowest in Block 1 in all three conditions, significantly increased between Blocks 1 and 2, and did not significantly increase between pairs of blocks beyond 1 and 2. Speed in Block 5 (in which subjects flew

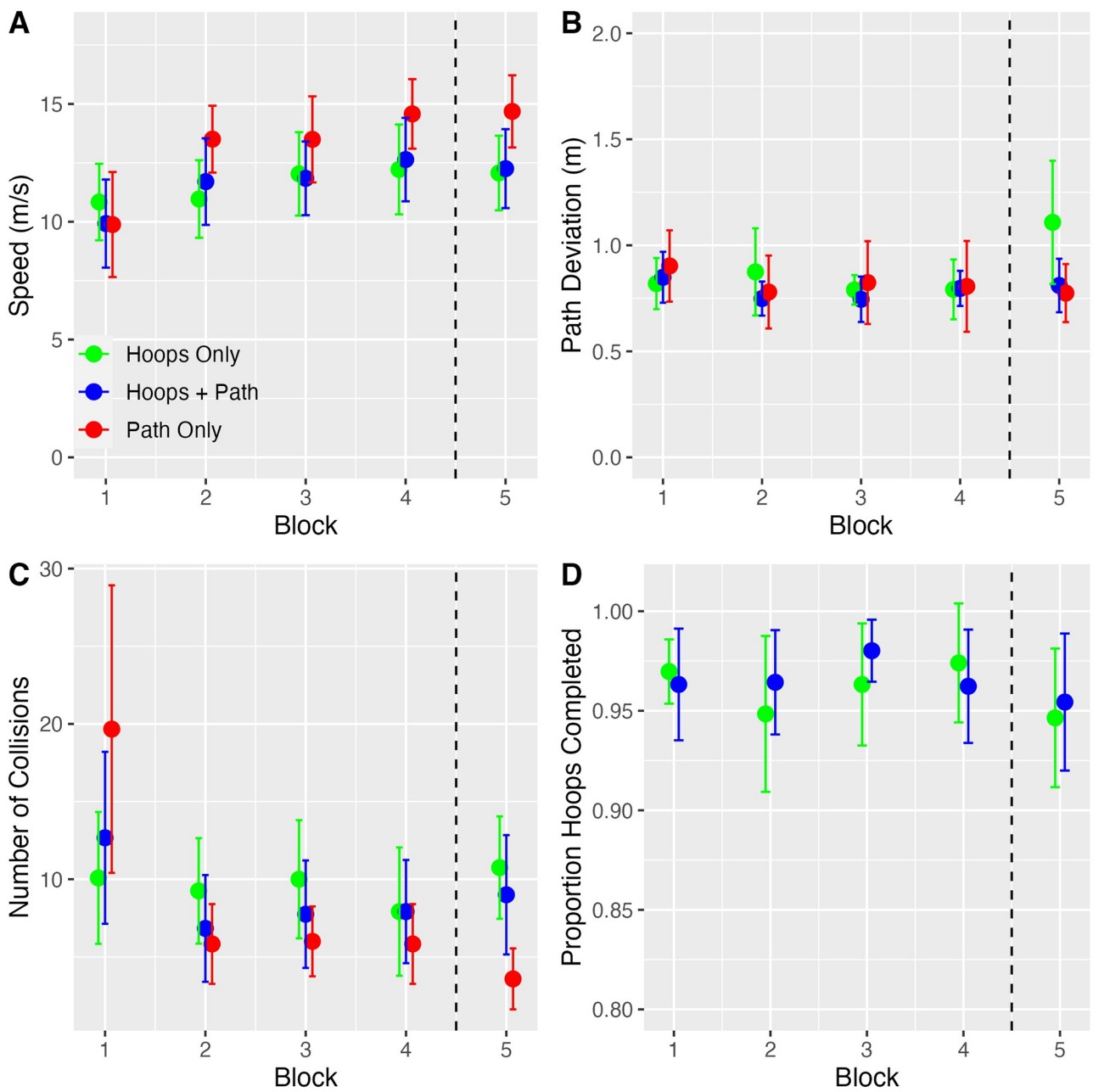

**Fig 2. Analyses of performance measures in Experiment 1: Mean speed (A), path deviation (B), number of collisions (C), and proportion of hoops completed (D).** Each measure is plotted as a function of block number, with the different colors representing different conditions. Error bars represent 95% CIs.

around the path in the opposite direction) was not significantly different from Block 4. Subjects also flew significantly faster in the Path Only condition compared to the other two conditions, although the effect sizes were moderate (d = 0.53 and d = 0.55).

Fig 2B depicts mean deviation from the center of the path broken down by condition and block. The mean overall path deviation was 0.83 m. For reference, the width of the path was approximately 1 m, which means that subjects remained on or close to the path for much of

the time. While it may seem surprising that path deviation was similar in the Hoops Only condition, when the path was not rendered, it is worth noting that the hoops were positioned on the path. Neither the main effect of condition ($F < 1$) nor the effect of block ($F_{4,20} = 2.70$, p = .06, $\eta^2 = 0.35$ [0.00, 1.00]) was significant, but the condition x block interaction was ($F_{8, 40} = 2.77$, p < .05, $\eta^2 = 0.36$ [0.05, 1.00]). Looking at Fig 2B, the source of the interaction appears to be the elevated path deviation in Block 5 of the Hoop Only condition. Closer inspection of the individual subject data revealed that two of the six subjects had substantially greater path deviations in this condition x block combination. Upon further inspection, we noticed that there was at least one turn where it took longer for the upcoming hoop to become visible when traveling in the opposite direction due to occlusion by trees. In the absence of a visible path (i.e., in the Hoop Only condition), certain subjects may have strayed a bit farther before detecting the upcoming hoop. This could explain why path deviation was slightly higher in Hoop Only condition when the loop was completed in the opposite direction.

The number of collisions per lap (Fig 2C) was greatest in Block 1 in all three conditions, decreased in Block 2, and remained relatively stable between 2 and 5. The main effect of condition was not significant ($F < 1$), nor was the main effect of block ($F_{1.11, 5.53} = 4.61$, p = .07, $\eta^2 = 0.48$ [0.12, 1.00]), but the condition x block interaction ($F_{1.76, 8.79} = 5.09$, p < .05, $\eta^2 = 0.50$ [0.24, 1.00]) was significant.

The mean overall proportion of hoops completed (Fig 2D) was 0.955. Neither of the main effects ($F_{1,5} = 1.41$, p = .29, $\eta^2 = 0.12$ [0.00, 1.00] for condition, $F < 1$ for block) nor the interaction ($F_{4, 20} = 1.17$, p = .36, $\eta^2 = 0.19$ [0.00, 1.00]) were significant. If we use a stricter definition of success that only includes hoops that were completed without the quadcopter colliding with or grazing the edge of the hoop before passing through, the overall proportion drops but only to 0.847. In other words, subjects cleanly passed through the large majority of hoops. Taken together, this suggests that subjects were able to successfully fly through a high percentage of hoops right from the beginning of the experiment regardless of whether the path was present or absent.

To summarize, the analyses of performance measures revealed that subjects performed the task well. They flew at a fast speed, successfully followed the path and passed through a high percentage of hoops, and infrequently collided with objects. Performance tended to be worse in Block 1 but improved and remained relatively stable beyond that block, including Block 5 in which subjects travelled around the loop in the opposite direction. These trends were present across all three conditions, with small differences emerging in just a few cases.

In each of the analyses above, all five blocks were included even though subjects flew around the loop in the opposite direction in Block 5. One might feel that the condition x block interaction should be evaluated without Block 5 included. We re-ran the analyses without Block 5 and found that none of the interaction effects changed from significant to non-significant or vice-versa, with one exception. In the analysis of path deviation, the condition x block interaction was not significant when Block 5 was excluded ($F_{6, 30} = 0.79$, p = .58, $\eta^2 = 0.14$ [0.00, 1.00]).

## Did subjects cut the corner on sharp turns?

In assessing the hypothesis that humans look where they want to go, one cannot assume that subjects always intended to follow the path. Although subjects were instructed to follow the path as closely as possible, they were also told to maintain a high speed. As such, it remains possible that subjects attempted to "cut the corner" on sharp turns. If they did, that would complicate the analysis of gaze behavior since fixating points that lie off the path while cutting the corner would not necessarily be inconsistent with the general hypothesis that people look where they want to go and steer in the direction of gaze [16].

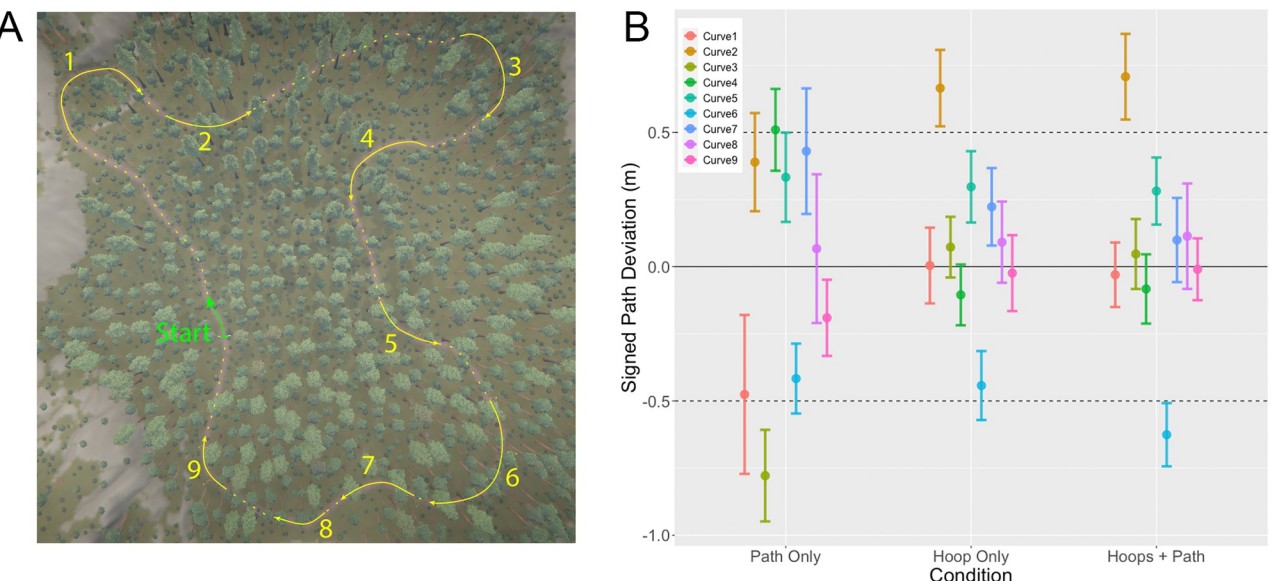

**Fig 3.** (A) Bird's eye view of virtual environment with the path highlighted by the dashed yellow line. The solid yellow lines indicate the curved sections of the path that were used in the analysis of signed path deviation. Curves are numbered 1 through 9. (B) Signed path deviation for each numbered section of the path in the Path Only, Hoop Only, and Hoops + Path conditions. Positive values correspond to an outside bias and negative values correspond to an inside bias.

To determine whether subjects exhibited corner-cutting behavior, we identified nine individual segments of the loop where the path was curved (see Fig 3A) and measured the signed deviation from the path center at each time step within those segments (see Fig 3B). A positive path deviation indicates an outside bias, and a negative path deviation indicates an inside bias. In the Path Only condition, there was an inside bias on four of the nine curves (1, 3, 6, and 9) but the 95% CI overlapped with the inside edge of the path (-0.5 m) for all but one curve (Curve #3). In addition, signed path deviation was positive (indicating an outside bias) on just as many curves (2, 4, 5, and 7). This is not surprising because the path was surrounded by trees on either side, so it was difficult or impossible on most curves to cut the corner without risking a collision. Signed path deviation was even closer to zero in the Hoops Only and Hoops + Path conditions. This makes sense because the hoops were positioned on the path, so in general, subjects could not cut the corner without missing a hoop. Taken together, this analysis rules out the possibility that subjects aggressively cut the corner when traveling around sharp bends in the path.

## Where did subjects look?

Next, we consider how much time subjects spent looking at each of the following categories of objects: the path, the edge of the hoop, through the center of the hoop, nearby trees, and the background. Nearby trees were defined as those that fell within a distance of 34.4 m of the subject, which corresponds to two times the average distance between hoops. We chose this distance because we wanted the proportion of time spent fixating nearby trees to reflect fixations to objects that subjects may be attempting to avoid at that moment. We reasoned that if gaze fell on trees that were more than two hoops ahead, it was unlikely to be for the purposes of obstacle avoidance. The background included the ground surface other than the path, trees other than those that were classified as nearby, and the sky.

In the Path Only condition, gaze was aimed at the path on only about half of the frames (49%). Equally often, subjects looked at either nearby trees (20%) or the surrounding terrain (29%) (see Fig 4A). Further analyses revealed that when subjects were not looking at the path, they were looking near the path. We quantified this by calculating for each frame on which gaze fell on the ground the distance between the point of fixation in the 3D environment and the center of the path. Fig 4B shows the density of fixations as a function of distance from the center of the path. The median distance was 1.11 m, which was slightly greater than the width of the path (approximately 1 m), suggesting that the majority of fixations were either on or near the path. Thus, in the Path Only condition, subjects only looked at the path itself 49% of

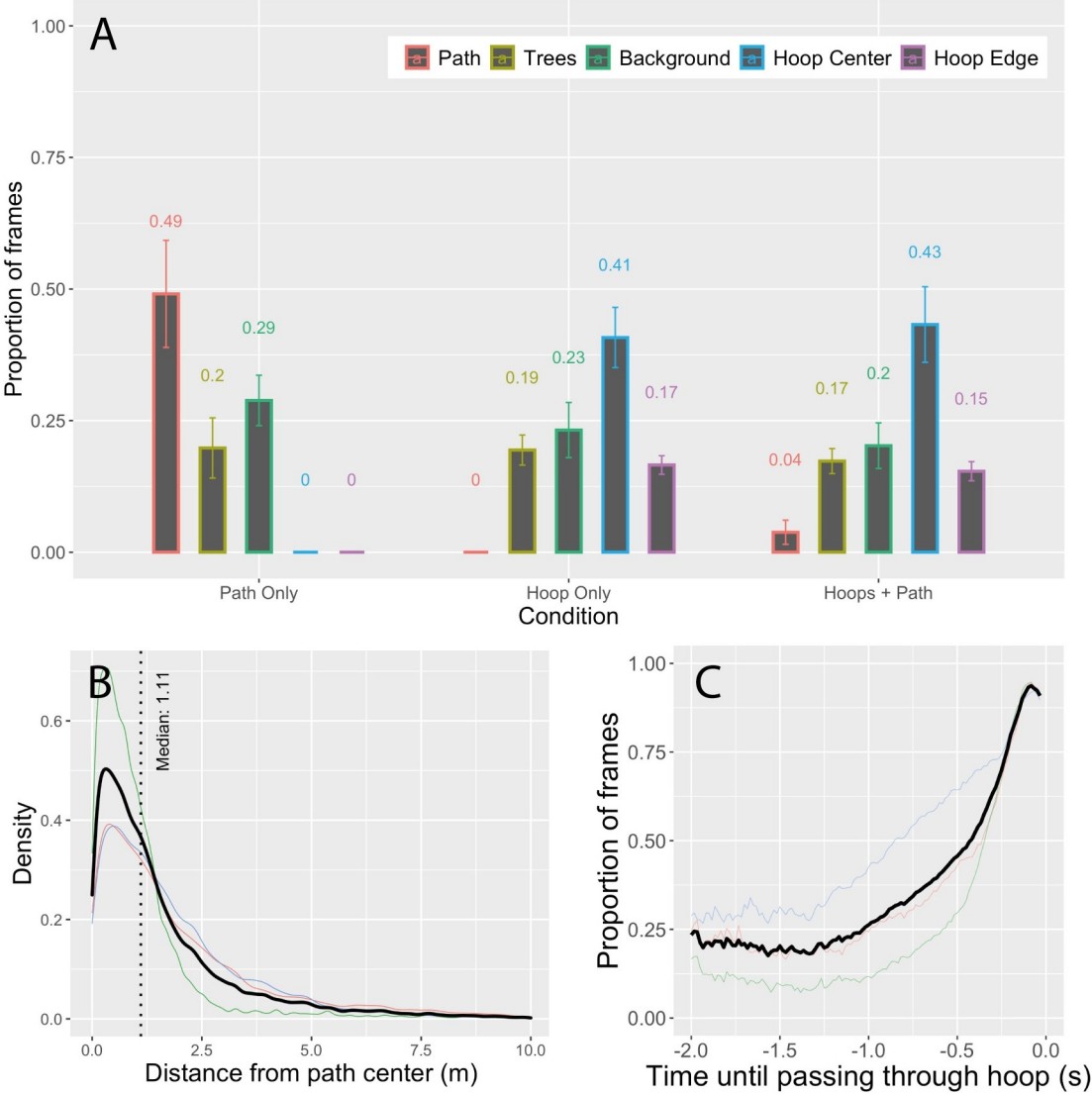

**Fig 4.** (A) Proportion of frames spent looking at the path, nearby trees, background, hoop center, and hoop edge in the Path Only, Hoops Only, and Hoops + Path conditions of Experiment 1. (B) Density plot of distance from fixation point to center of path. (C) Proportion of frames spent looking through the center of the hoop as a function of the amount of time until passing through the hoop. In B and C, the thin colored curves represent individual subject data and the thicker black curve represents the mean across subjects.

the time and gaze often fell elsewhere (on trees and nearby terrain) but subjects spent most of their time looking in close proximity to the path.

In the Hoops Only condition, the desired future path travels through the center of each hoop, which is empty space. As such, there were no visible objects on the desired future path to anchor gaze as there was when following a path. The things that were nearest to the desired future path were the edges of the hoops, but those were fixated only 17% of the time (see Fig 4A). The object category that drew the highest percentage of fixations (41%) was the hoop center, indicating that subjects spent more time looking at distant objects that were visible through the center of the hoop than they did at any other object category.

Before we can conclude that looking through the hoop center was a deliberate gaze strategy, it is necessary to rule out an alternative explanation for the observed results. As subjects approached a hoop, the center of the hoop took up an increasingly larger part of the visual field. Eventually, there was no place to look other than through the center of the hoop. Nevertheless, the hoop did not occupy the majority of the visual field until the last few frames. Fig 4C shows the proportion of frames spent looking through the hoop center as a function of the amount of time remaining until passing through the hoop, cut off at -2 s because the median time between hoops was 1.6 s. At -1 s, the largest visual angle that the hoop could occupy (assuming a head-on approach to the hoop at maximum speed) was approximately 5 deg of visual angle along the horizontal or vertical axis. Yet subjects were looking through the hoop center ~26% of the time. Even at -¼ s, when subjects were looking through the hoop about 70% of the time, the maximum visual angle that the hoop could occupy was ~19 deg. Hence, looking through the center of the hoop does not appear to be a necessary consequence of steering through the hoop but rather part of the gaze strategy for guiding steering.

In addition, gaze fell on trees 19% of the time and on background scenery 23% of the time. Such fixations may have occurred when subjects were searching the scene for upcoming hoops.

Lastly, gaze behavior in the Hoops + Path condition was very similar to the Hoops Only condition (Fig 4A). This is interesting because the path was visible in this condition, yet subjects spent very little time (~4% of frames) looking at it. Recall from the analyses of performance measures (average speed, path deviation, proportion of hoops completed, and number of collisions) that the differences between the Hoops Only and Hoops + Path condition were negligible. Thus, when the task involved steering through waypoints (i.e., hoops), the presence of the path had minimal impact on gaze behavior and task performance.

To summarize, gaze often fell on things that were not the path or the center of the upcoming hoop. This was almost certainly due to the presence of dense clutter, which meant that the upcoming path and the upcoming hoops did not always pop-out—they had to be searched for. (It is also possible that the estimated proportion of fixations to the path and upcoming hoops was reduced due to error in the measurement of gaze direction. We return to this point in the Discussion.) Nevertheless, subjects often looked in the general direction of things that they wanted to move toward—the future path or through the center of the upcoming hoop. Second, subjects performed the task extremely well. They moved at ~12 m/s, stayed within less than 1 m of the center of the path, passed through more than 95% of the hoops, and avoided collisions. Lastly, gaze was aimed at trees only about 25% of the time. By itself, this does not tell us much about whether people need to fixate and track obstacles to avoid them. As such, we turn to the results of Experiment 2 in which the density of trees was manipulated.

The proportion of time spent looking at trees in Experiment 2 was quite low–only about 18% and 7% in the Dense Tree and Sparse Tree conditions, respectively (Fig 5). This is actually lower than it was in Experiment 1, although the small sample size prohibits statistical comparisons. Thus, despite the large increase in the density of trees near and on the path, the amount

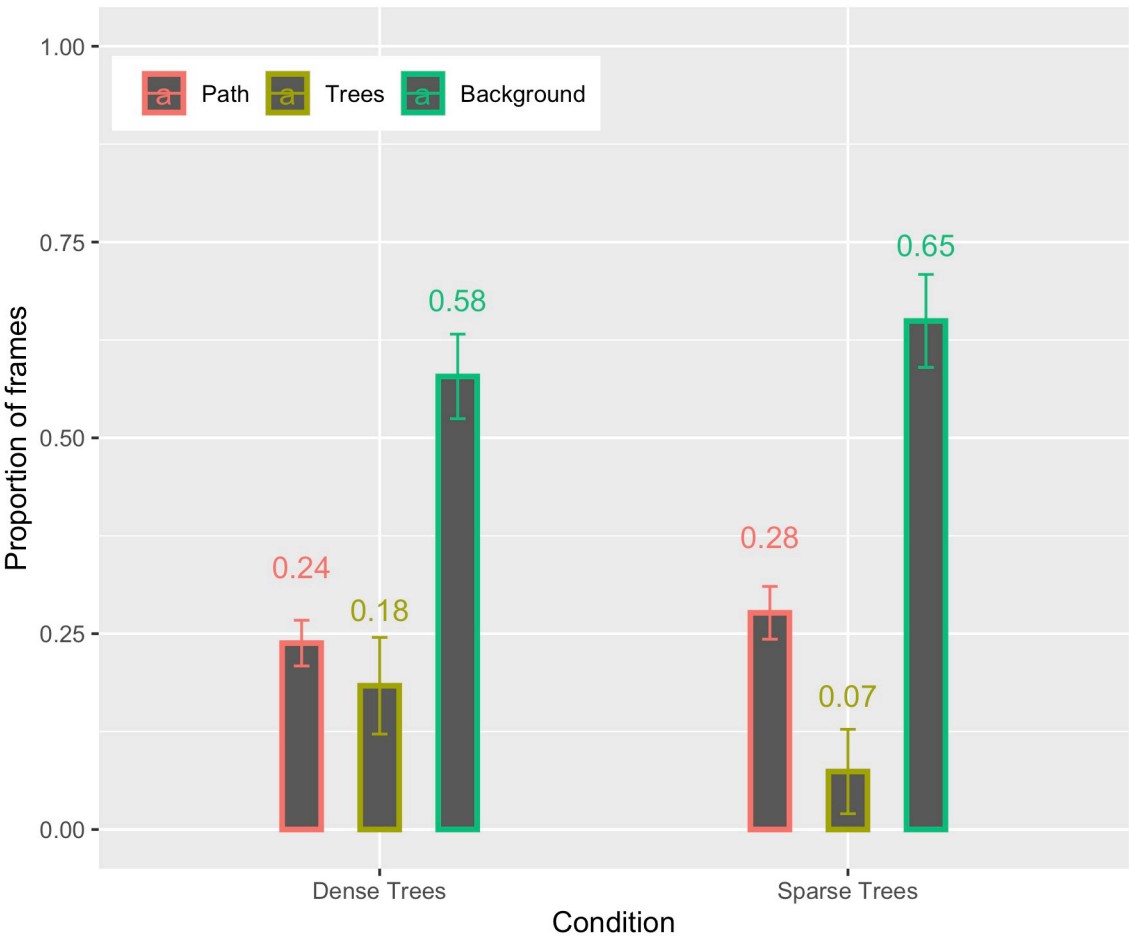

**Fig 5. Proportion of frames spent looking at the path, nearby trees, and background in the Dense Trees and Sparse Trees conditions of Experiment 2.**

of time spent looking at trees did not increase. This suggests that subjects can successfully avoid collisions with obstacles near their future path without sustained tracking of those objects. S1 Movie is from a representative subject in the Dense Tree condition of Experiment 2 and illustrates the basic findings from that condition: Gaze fell near but not always on the path and there were infrequent fixations on obstacles (the trees). Taken together, the findings support the hypothesis that a critical role for gaze during high-speed navigation in cluttered environments is to search for the path. Humans are capable of steering through waypoints and following a path at high speeds even without persistent fixation of these parts of the scene.

## Did subjects make anticipatory steering adjustments?

The second aim of this study was to determine whether subjects made anticipatory steering adjustments. Whereas evidence for look-ahead fixations has been found in previous studies, the evidence for anticipatory steering adjustments is less clear. Next, we explain how we went about testing for anticipation. If subjects adapted their approach to the upcoming hoop (Hoop N) in anticipation of the subsequent hoop (Hoop N+1), then their approach trajectory to hoop N should depend on the position or perhaps orientation of N+1 relative to N. That is, the

approach trajectory that they take when hoop N+1 is in one location should be different than when hoop N+1 is in a different location (see Fig 6A). In other words, the relative position of hoop N+1 should be a significant predictor of the approach angle to Hoop N.

This prediction can be expressed in the form of a linear model with approach angle ($\alpha_N$) to Hoop N as the outcome variable and angular position of Hoop N+1 relative to Hoop N ($\theta_{N+1,N}$) as the predictor. Both angles are defined relative to the normal vector of Hoop

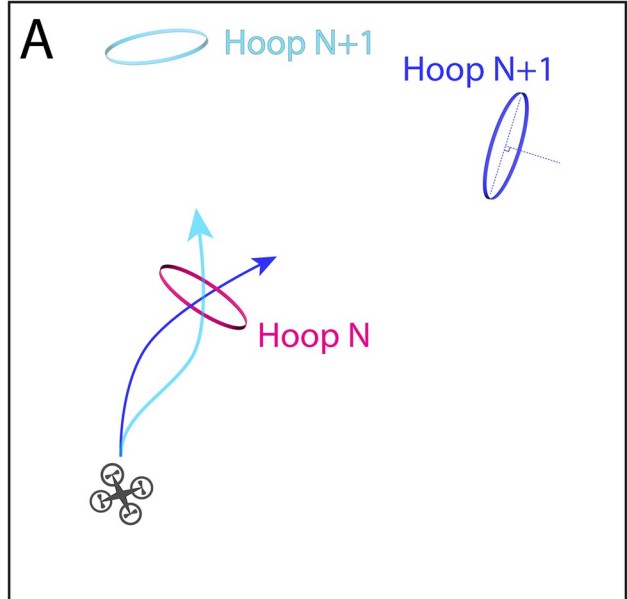

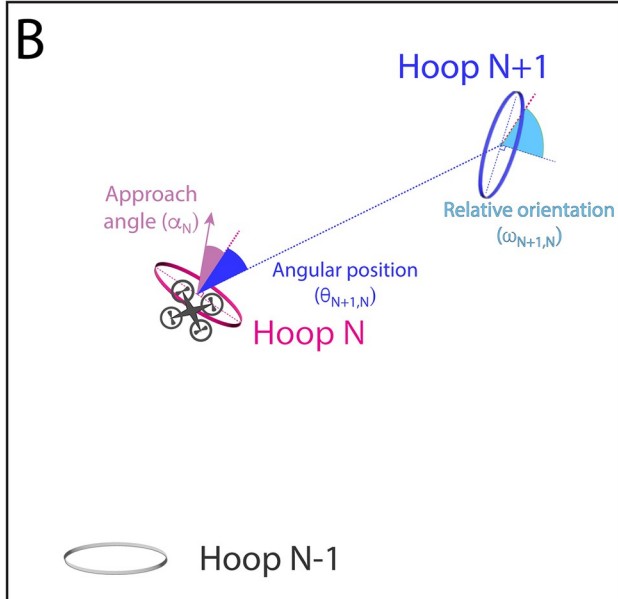

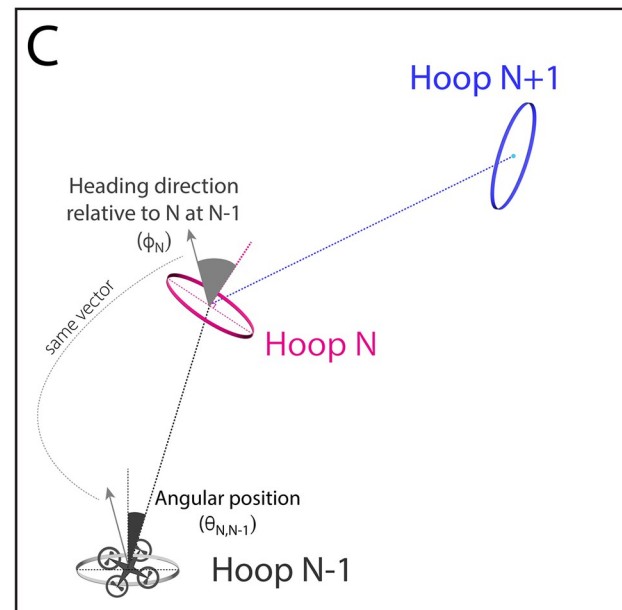

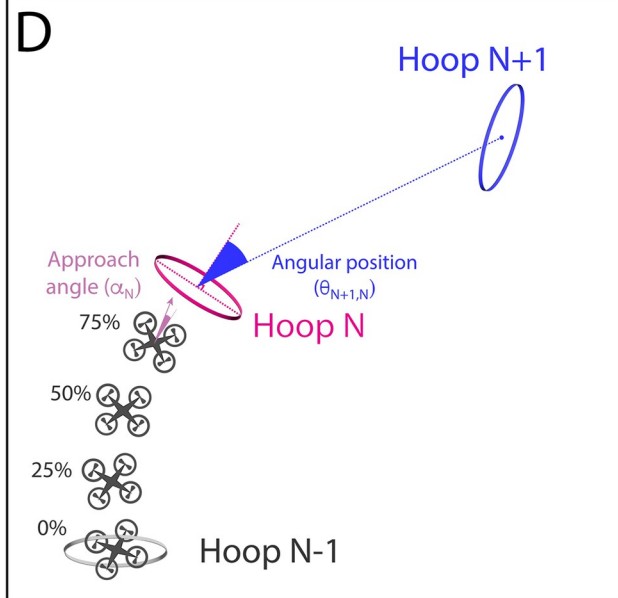

**Fig 6.** (A) Illustration of how the trajectory to the most immediate hoop (Hoop N) may depend on position of subsequent hoop (Hoop N+1). (B) Predictor variables (angular position and relative orientation), outcome variable (approach angle), and (C) covariates (heading direction relative to N at N-1, angular position of N relative to N-1) used in linear model. (D) Illustration of outcome variable (approach angle) measured at different positions leading up to Hoop N.

N (see Fig 6B). The approach angle to hoop N may also depend on the initial conditions for the segment of the path between hoop N-1 and hoop N, including the drone's heading direction relative to hoop N as it passes through hoop N-1 ($\phi_N$), the drone's orientation relative to hoop N as it passes through hoop N-1 ($\rho_N$), and the angular position of hoop N relative to hoop N-1 ($\theta_{N,N-1}$) (Fig 6C). Hence, we included these variables as covariates in the model, which was a linear mixed-effects model with subjects as a random factor:

$$\alpha_N \sim \phi_N + \rho_N + \theta_{N,N-1} + \theta_{N+1,N} \tag{1}$$

Fitting this model to the data, we found that the angular position of Hoop N+1 accounted for 23.6% of the variance in approach angle (measured on the last frame before passing through hoop N) that was left unexplained by the covariates. We also compared the model in Eq 1 to another model with the covariates alone (i.e., without $\theta_{N+1,N}$). The likelihood ratio test was statistically significant ($\chi^2(1) = 413.70$, $p < .001$, dAIC = -412), indicating that angular position of Hoop N+1 was a significant predictor of approach angle to Hoop N. This is consistent with a strategy that involves anticipation of upcoming steering constraints.

Next, we asked how far in advance of reaching Hoop N did subjects start to anticipate Hoop N+1. We repeated the analysis with the outcome variable (approach angle) measured at four additional points in time: when the subject was 75%, 50%, 25% and 0% of the way to reaching Hoop N (Fig 6D). Not surprisingly, the proportion of variance explained decreased as approach angle was measured farther back in time (Fig 7A). Nevertheless, there was still some evidence of anticipation early on in the segment.

Lastly, we fit the model in Eq 1 to the data from individual subjects. The variation in partial $R^2$ values (see Fig 7A) suggests that some subjects anticipated upcoming hoops to a greater degree than others, especially toward the end of each segment (i.e., at 100%). However, all six subjects exhibited evidence of anticipation.

This analysis suggests that subjects took the relative position of the upcoming hoop into account. Next, we consider whether they also took relative orientation into account. In this analysis, the main predictor is the orientation of hoop N+1 relative to hoop N ($\omega_{N+1,N}$ in Fig 6B). We found that only a small proportion of the variance in approach angle was explained by relative orientation (Fig 7B), suggesting that subjects did not adapt their approach trajectory to Hoop N in anticipation of the relative orientation of Hoop N+1.

## Discussion

In this section, we summarize the main findings and expand upon their broader significance for our understanding of the gaze and control strategies for high-speed steering through cluttered environments. Our summary is organized into six key findings.

First, in the conditions that required subjects to steer through hoops, the predominant gaze behavior was to look through the center of the upcoming hoop. This is consistent with the findings of [20], which is the one previous study on gaze behavior in drone pilots. However, in the Hoops Only condition of the present study, subjects also spent a significant portion of time looking elsewhere (e.g., at nearby trees and surrounding terrain), which was not reported by [20]. Such differences can be attributed to the additional complexity of the scene used in the present study compared to the relative sparse environment used in [20], which lacked obstacles other than hoop edges and had less complex background scenery. The presence of clutter in the present study meant that subjects had to actively search for upcoming hoops by scanning the scene.

Second, in the conditions that required following the path, subjects performed the task quite well despite the fact that the proportion of frames spent fixating the path was only about

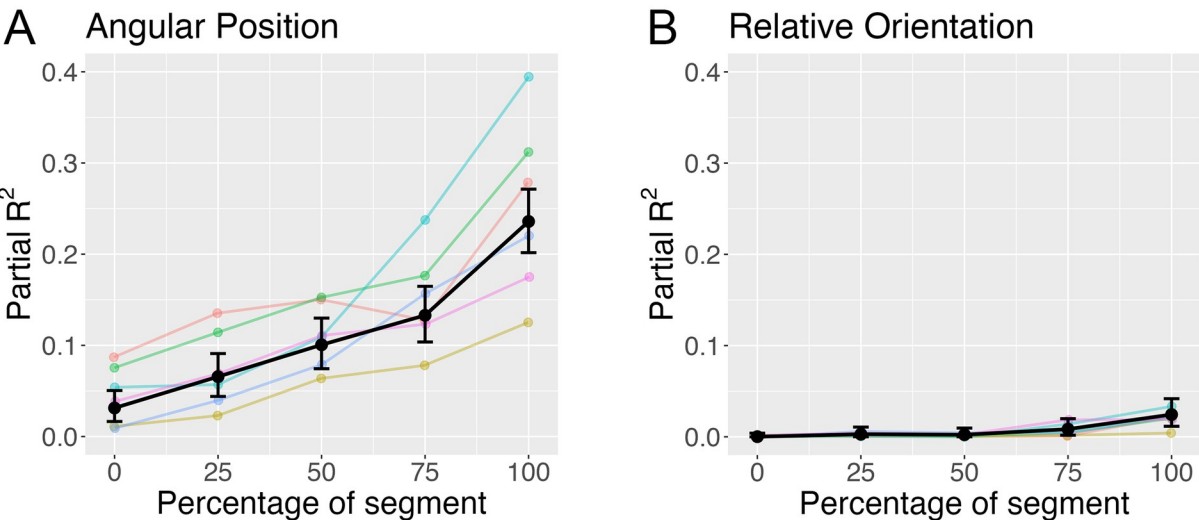

**Fig 7. Partial $R^2$ as a function of the percentage of segment with angular position (A) and relative orientation (B) as the predictor variable.** Colored curves show partial $R^2$ based on individual subject data.

0.25 to 0.50. As we acknowledge in the Limitations section below, it is possible that this range is deflated due to error in measurement of gaze direction. However, the presence of dense clutter ensured that gaze fell off the path on a substantial proportion of frames. This suggests that sustained fixation of points on the future path is not necessary to successfully control steering. As mentioned in the introduction, many theories and models of steering assume that the information that is needed to skillfully control steering is revealed by fixating points on the ground over which the actor intends to travel. Given the results of the present study, an important goal for future work is to explain how humans are able to successfully negotiate a winding path when gaze often falls off the path. One possibility is that steering is coupled to gaze but only when the gaze point falls on the desired future path. In our view, a more likely possibility is that the information that is relevant for controlling steering can also be picked up using peripheral vision. If so, precise fixation of points on the future path may not be as critical as previously assumed.

Third, although subjects spent less than half the time looking at the path, gaze often fell near the path in the conditions that required following the path. More generally, subjects were able to direct gaze at or near the things that they were instructed to follow or move toward. This is interesting because subjects were moving at such high speeds, and because the path and the hoops blend into the background scenery, and were often hidden in shadows, or occluded by trees and hills. This points to a significant gap in our understanding of the role of gaze in steering [4]. Existing theories and models attempt to capture how steering is regulated based on currently available information, which is assumed to be made available by looking at the future path. However, because the focus has been on steering in uncluttered environments, the question of how humans know where to look to find the future path has received much less attention.

Fourth, pilots did not often direct gaze at things that they wanted to avoid, such as trees and the edges of the hoops. This suggests that obstacle avoidance does not require sustained tracking. A tentative conclusion is that the information needed to guide obstacle avoidance can be detected using peripheral vision. It is worth noting, however, that all the obstacles in this study were stationary. Avoidance of moving obstacles may require sustained tracking [25].

Fifth, pilots made anticipatory steering adjustments; that is, they adapted how they approached the upcoming waypoint in anticipation of having to steer through future waypoints. As explained in the introduction, evidence of anticipatory steering adjustments in previous studies is inconsistent. The fact that such behavior was exhibited in this study could reflect the presence of significant inertial constraints or the high level of piloting skill of the subjects. Future research should explore the conditions in which actors do and do not exhibit anticipatory steering adjustments, the role of skill, and the control strategies that capture such behavior. It is common in the literature to assume that anticipatory steering behavior reflects a path planning process that relies on internal forward and inverse models [4]. In principle, however, such behavior could also emerge from the coupling of information and action according to some control law [26].

Lastly, we found no significant differences on any performance measure on Block 5 (in which subjects traveled around the loop in the counterclockwise direction) compared to Block 4. This suggests a lack of evidence that subjects' knowledge of the spatial layout of the environment played a role in their ability to perform the task. It could be that four blocks of trials were not sufficient for subjects to learn the course well enough for that knowledge to be useful to guide steering. Alternatively, it is possible that actors rely primarily on currently available information rather than knowledge of spatial layout. Further research is needed to tease apart these possibilities.

## Implications for control of automobile steering

On the surface, it may seem that the findings of the present study are of little relevance to the visual control of steering along a winding road. After all, steering a car along a winding road differs from flying a quadcopter along a winding path in a forest in several respects. The controller dynamics differ, elevation is fixed in a car but variable in a quadcopter, and many of the sources of clutter that occluded the path in the present study (e.g., branches) are not found on roads. Given the absence of clutter on roadways, one could argue that the problem of searching for the desired future path is not an issue in the context of automobile driving as it was in the present study. While that may be true when driving under ideal conditions, people also frequently drive under less idealized conditions in which the upcoming road may be hard to distinguish from the surrounding terrain due to poor visibility. For example, when driving at night, in fog, and in heavy rain or snow, the road does not always immediately pop out. This is especially true if the lane markers are faded or if the driver has poor vision. As such, the problem of searching for the upcoming road is an important component of automobile driving in certain scenarios. Identifying the strategies that drivers use to search for the road in such conditions should be an important goal for future research. Likewise, it would be useful to expand models of steering to better account for human steering performance under conditions in which sustained fixation of the road is not possible. Such models could have useful applications for predicting driver performance under conditions in which visibility is degraded, which is an important but less well understood area of driver behavior modeling. By improving models of driver behavior, engineers could conduct more accurate offline virtual safety evaluations of driver assistance systems [27].

## Limitations

Caution must be exercised when generalizing the findings due to the small sample size and the focus on subjects with extensive FPV drone piloting experience. For example, subjects in the present study were able to perform the task well despite the presence of dense clutter, which prevented continual fixation of the upcoming path. Nevertheless, it is possible that for less

skilled actors, successful steering depends more critically on the ability to continually fixate the future path. Likewise, there may be individual differences or differences across skill levels in anticipatory steering behavior.

Another reason for caution stems from error in the measurement of gaze direction from the eye tracker. We attempted to mitigate the impact of error by focusing our analyses of gaze behavior on the three subjects whose mean eye tracking error during the calibration phase was less than 5 degrees overall and 2 to 3 degrees in the center of the display. It remains possible that the actual proportion of time spent looking at the upcoming path, the hoop edges, or the hoop center was greater than what was found in our analyses. However, given the speed at which subjects were moving and the fact that hoops and sections of the path were often occluded by trees, branches, or small hills or hidden in shadows, it is highly likely that gaze fell on the background or nearby trees or terrain on a substantial proportion of frames.

## Conclusions

Although gaze and steering appear to be tightly coupled when driving along a winding road on a flat, obstacle-free ground surface, the control of steering is not likely to be as directly linked to gaze direction when moving through cluttered environments. The need to search for navigable spaces within the clutter means that some fixations will fall on objects and surfaces that are not in the intended direction of locomotion. Furthermore, humans adapt how they approach the most immediate waypoint in anticipation of future waypoints, suggesting that non-fixated waypoints also influence steering. It follows that the control of steering is not always guided by the direction of gaze and that sustained fixation of the desired future path is not necessary for following paths, steering through waypoints, and avoiding obstacles. Either intermittent fixation of the future path is sufficient, or actors can also rely on information picked up by peripheral vision.

## Supporting information

**S1 Movie. Recording of a sample subject from the Dense Trees condition of Experiment 2, in which subjects were instructed to follow the path while avoiding trees and overhanging branches.** The red crosshair shows the point of fixation.
(MP4)

## Author Contributions

**Conceptualization:** Nathaniel V. Powell, Xavier Marshall, Brett R. Fajen.

**Data curation:** Nathaniel V. Powell.

**Formal analysis:** Nathaniel V. Powell, Gabriel J. Diaz, Brett R. Fajen.

**Funding acquisition:** Brett R. Fajen.

**Investigation:** Nathaniel V. Powell, Brett R. Fajen.

**Methodology:** Nathaniel V. Powell, Gabriel J. Diaz, Brett R. Fajen.

**Project administration:** Brett R. Fajen.

**Resources:** Brett R. Fajen.

**Software:** Nathaniel V. Powell, Xavier Marshall, Brett R. Fajen.

**Supervision:** Brett R. Fajen.

**Validation:** Nathaniel V. Powell, Gabriel J. Diaz, Brett R. Fajen.

**Visualization:** Nathaniel V. Powell, Xavier Marshall, Brett R. Fajen.

**Writing – original draft:** Nathaniel V. Powell, Brett R. Fajen.

**Writing – review & editing:** Nathaniel V. Powell, Xavier Marshall, Gabriel J. Diaz, Brett R. Fajen.

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
