## [Decision Letter · Decision Letter 0]

28 Sep 2023

PONE-D-23-23553Coordination of gaze and action during high-speed steering and obstacle avoidancePLOS ONE

Dear Dr. Fajen,

Thank you for submitting your manuscript to PLOS ONE. After careful consideration, we feel that it has merit but does not fully meet PLOS ONE’s publication criteria as it currently stands. Therefore, we invite you to submit a revised version of the manuscript that addresses the points raised during the review process.

We look forward to receiving your revised manuscript.

Kind regards,

Markus Lappe

Academic Editor

PLOS ONE

4. We note that Figure 1 in your submission contain copyrighted images. All PLOS content is published under the Creative Commons Attribution License (CC BY 4.0), which means that the manuscript, images, and Supporting Information files will be freely available online, and any third party is permitted to access, download, copy, distribute, and use these materials in any way, even commercially, with proper attribution. For more information, see our copyright guidelines: http://journals.plos.org/plosone/s/licenses-and-copyright.

Reviewers' comments:

Reviewer's Responses to Questions

**Comments to the Author**

1. Is the manuscript technically sound, and do the data support the conclusions?

Reviewer #1: Partly

Reviewer #2: Partly

2. Has the statistical analysis been performed appropriately and rigorously? 

Reviewer #1: I Don't Know

Reviewer #2: Yes

3. Have the authors made all data underlying the findings in their manuscript fully available?

Reviewer #1: Yes

Reviewer #2: Yes

4. Is the manuscript presented in an intelligible fashion and written in standard English?

Reviewer #1: Yes

Reviewer #2: Yes

5. Review Comments to the Author

Reviewer #1: The study explores gaze and steering behavior of drone pilots engaged in first person-view drone racing within a dense forest environment. While participants effectively completed the racing tasks, they looked less to their future waypoints than expected. Participants anticipated upcoming hoops, optimizing their steering accordingly. Overall, the paper is well-written and provides valuable insights into human navigation behavior in FPV drone racing.

My main issue revolves around statistics. I am worried about the presentation of the statistical analysis.

First, the sample size in Experiment 1, involving only six subjects, and particularly the inclusion of only three subjects in Experiment 2 (one of them an author), raises concerns. PLOS One places great emphasis on appropriate sample sizes. Given the requirement for drone piloting expertise, it's understandable that recruiting a larger sample was challenging. Now that the pandemic has eased, it may be worthwhile to explore the possibility of obtaining additional subjects to enhance the robustness of the study. But I realize this might not be possible. For now, the decision to primarily focus on the results of Experiment 1 appears justified.

Second, the statistical details in the paper require further clarification. For instance, in line 316, it would be helpful to specify which post-hoc tests were employed and whether any correction method for multiple comparisons was applied. Additionally, when discussing the repeated-measures ANOVA, it should be explicitly mentioned that "condition" was treated as a between-subject factor.

The inverted block five really stretches the definition of repeated measures. However, it doesn't seem to be statistically different from the other blocks, except for what the authors describe as a "spurious finding." Does the analysis yield consistent results when Block 5 is excluded, apart from this "spurious" interaction?

The absence of statistical tests in the "Where did subjects look?" section could be clarified. Is this related to the exclusion of subjects with imprecise eye-tracking data? For Experiment 2, the small sample size is mentioned in line 440. Were the three subjects excluded from the eye movement analysis the same three who did not participate in Experiment 2?

In the section on anticipatory steering adjustments, an LMM is used, but no specific model results are presented. It seems like the analysis involved adding the angular position of Hoop N+1 to a separate LMM and calculating an LRT. This procedure should be explicitly documented for clarity.

While the statistical methodology appears sound, the presentation would benefit from some refinement to ensure transparency and enable readers to have a clearer understanding of the results.

Apart from these issues, the discussion already addresses the problem of eye tracking precision, and the authors are best positioned to assess its implications for the study.

Notes:

1. In line 257, Experiment 2 is described as identical to Experiment 1, but it remains unclear to me which configuration of hoops and path from Experiment 1 was used.

2. The eye tracking errors (exceeding 5 degrees for excluded subjects and averaging 3.05 degrees for the remaining subjects) are notably large. While the standards for eye tracking in VR differ from stationary eye trackers, it could be beneficial to include some discussion regarding why these errors were this large.

3. This is a minor point, but in line 575, a comparison with car steering is drawn: While some arguments against this comparison are given, the absence of elevation in automobile steering could be acknowledged, and the possibility of testing this with a larger sample size could be pointed out. It might be helpful to clarify the intended message here?

4. The paper lacks mention of motion sickness, despite the nature of the task, which could typically lead to such issues. While the Vive Pro is a solid device, motion sickness can still occur. Was motion sickness measured during the study? Were any countermeasures taken beforehand to mitigate its effects on the participants?

5. Figures 3 and 4 suffer from excessive JPEG compression in the PDF version.

Reviewer #2: I read this paper with interest. The topic is timely and relevant, and the general approach is novel.

I do, however, have a number of concerns that would appear to weaken many of the conclusions of the paper.

The first set of concerns focus on the author acknowledged weaknesses. The limitations section of the paper identifies three aspects of the design that limit the study (sample size, participant expertise and error in gaze measures), however these limitations do not seem to be fully considered elsewhere. The abstract does not state the sample size, or acknowledge these weaknesses, and throughout the manuscript conclusions are drawn from the data as if these issues do not exist. In the analysis I would have been really interested to see more examination of the individual patterns of behaviour in both steering trajectories and gaze behaviours. Currently the only hint is provided in Figure 3B which shows the extent of variation in gaze performance for 2 of the 3 individuals (1 dataset is masked by the average line). Instead, the manuscript focusses on presenting group analysis for each measure which masks that this performance will be highly influenced by a single participant. Similarly the limitations section flags that eye-tracking measures had an error of 2-5deg, but this is not addressed in the eye-tracking analysis in terms of the likelihood of misclassification of target objects/regions or estimating gaze distance from path center.

The second set of concerns tie in with the novelty of the task used and the extent to which flying a drone in the manner specified in the experiment is comparable to the experiments referenced in the introduction looking at steering along paths.

The task given to participants was: “fly as quickly as possible while staying close to the path, and attempting to pass through each hoop (when present)”. This task explicitly prioritises speed over accuracy, and would appear to reduce the importance of spatial accuracy with respect to the rendered path and the hoop.

With these instructions I would expect trajectories to only loosely align to the rendered path since the priority is taking the quickest route (that passes through the hoops). The equivalent when driving would cause drivers to take the “racing line” which leads to ‘corner cutting’ (deviation away from the center of the path). This is crucial for interpreting the presented data because research shows that such instructions can alter gaze patterns – e.g Reference [16] showed that when drivers were instructed to take the racing line, gaze fell on the inside edge of the demarcated road.

The task instructions causes two problems for the manuscript in it’s current form: (A) It is unclear that the analysis of steering and gaze behaviours are sufficient/appropriate to identify whether the above behaviours occur, or to confirm whether gaze is being directed toward the paths actually taken (rather than the paths rendered on the screen); and (B) The conclusions that gaze is only weakly coupled with steering could simply be a result of the reduced importance/relevance of spatial information in this task.

I will expand in detail on some of the issues around (A) and (B):

It seems that the analysis of steering does not really show whether there is good compliance to the rendered path. The value of 0.83m deviation from path center is reported and used to suggest that the drone remained on or close to the path, however deviation of more than +/- 0.5m from the path center could be considered leaving the path (given the whole path is only 1m wide). The fact that path deviation metrics do not differ when the path is no longer rendered (and only hoops are present) would seem to suggest that the rendered path is only weakly influencing steering because it is not prioritised in the task instructions. Further examination of the shape of the trajectories relative to the path may be able to provide a better sense of whether the drone pilots were merely taking the shortest/quickest route through each lap (as instructed), or whether trajectories are in fact constrained by the rendered path.

The manuscript states in a few places that “subjects performed the task extremely well” however because the priority instruction was to travel quickly it makes interpreting spatial errors difficult because there will clearly be speed-accuracy trade-offs. Trials where there were large spatial errors (>10m left/right, or 20m vertically) were halted and a new trial was run instead, though no report of how often this occurred is given making it hard to interpret whether the data are representative of most flight attempts. Number of collisions reported in Figure 2 varies, but after very high values in the first block they seem to settle to approximately 5-10 per lap - the authors interpret this as “infrequent” but I find this value hard to interpret. Compared to driving a vehicle on a road (where no collisions are tolerated) this value still seems extraordinarily high, however, it is possible that a very low threshold is being used to classify a collision (i.e. is clipping a branch/leaf on a tree a collision?) or that this could be considered low given how many potential collisions with trees/hoops there might be during a lap.

Unfortunately, I could see no definition of what constitutes a collision in the manuscript, how long a lap takes (on average), nor was it clear whether anything happened to the drone as a result of a collision (i.e. did it just fly through the collision object, or get slowed down, or stopped?). Mention is made of the drone getting stuck in branches which suggests that some collision detection was used to alter movement of the drone. Related to this it is unclear whether the path/ground is itself a potential collision object – it seems likely it would need to be or the drone could pass through the ground, however this does then undermine a key conclusion that pilots did not need to look at collision obstacles. The fact that the ground could be considered an ‘attractor’ along 2 axes and a ‘repellor’ in the vertical axis would seem to be of potential interesting but I could not see any consideration of this characteristic (i.e. how high did the pilots fly above the ground and how consistent was this height maintained?). Depending on how collisions/spatial errors are handled, in terms of error feedback, you might expect different behaviours from the pilots – e.g. participants might be inclined to reduce spatial error/collisions if that led to slower trials, or if they knew for sure from error feedback that they had failed the task of passing through each hoop. The number of collisions approximately halved when hoops were removed (in block 5) so it seems possible that around half of collisions (~5) were with the hoops themselves, suggesting that pilots often clipped the hoop rather than passing through the centre. This also highlights an issue with the ‘hoops completed’ metric – what was the spatial cut-off for success? Did colliding with the hoop constitute ‘completion’? If there was no temporal penalty for collisions then it seem likely that trajectories will again be biased toward the quickest trajectory at the cost of passing cleanly through the hoops, which again would seem to weaken the importance of gaze and steering being tightly coupled.

The currently the reported metrics would seem to reinforce the idea that the drone pilots were accepting spatial trajectory errors as a trade-off for increasing their speed completing laps. This is important since it could explain why gaze does not appear to be directed to the rendered path or hoop as much as would be expected by the literature.

In summary whilst the experimental design appears to have the potential for revealing some interesting findings, in it's current form this manuscript does not provide sufficient evidence/justification to support the claims made.

6. PLOS authors have the option to publish the peer review history of their article (what does this mean?). If published, this will include your full peer review and any attached files.

Reviewer #1: No

Reviewer #2: **Yes: **Richard Wilkie

---

## [Author Response · Author response to Decision Letter 0]

5 Dec 2023

Our responses to the reviewers' comments are included in the "Response to Reviewers" document.

---

## [Decision Letter · Decision Letter 1]

16 Jan 2024

PONE-D-23-23553R1Coordination of gaze and action during high-speed steering and obstacle avoidancePLOS ONE

Dear Dr. Fajen,

Thank you for submitting your manuscript to PLOS ONE. After careful consideration, we feel that it has merit but does not fully meet PLOS ONE’s publication criteria as it currently stands. Therefore, we invite you to submit a revised version of the manuscript that addresses the points raised during the review process.

We look forward to receiving your revised manuscript.

Kind regards,

Lei Zhang, PhD

Academic Editor

PLOS ONE

Journal Requirements:

**Additional Editor Comments:**

The revised manuscript has largely improved. But there are still some places that need further revisions.

Reviewers' comments:

Reviewer's Responses to Questions

**Comments to the Author**

1. If the authors have adequately addressed your comments raised in a previous round of review and you feel that this manuscript is now acceptable for publication, you may indicate that here to bypass the “Comments to the Author” section, enter your conflict of interest statement in the “Confidential to Editor” section, and submit your "Accept" recommendation.

Reviewer #1: All comments have been addressed

Reviewer #3: All comments have been addressed

2. Is the manuscript technically sound, and do the data support the conclusions?

Reviewer #1: Partly

Reviewer #3: Partly

3. Has the statistical analysis been performed appropriately and rigorously? 

Reviewer #1: Yes

Reviewer #3: Yes

4. Have the authors made all data underlying the findings in their manuscript fully available?

Reviewer #1: Yes

Reviewer #3: Yes

5. Is the manuscript presented in an intelligible fashion and written in standard English?

Reviewer #1: Yes

Reviewer #3: Yes

6. Review Comments to the Author

Reviewer #1: The latest changes in the paper show a more careful consideration of limitations, a more transparent statistical analysis, and a discussion of "corner cutting behavior." The revised Anova is better explained, addressing assumptions and factors appropriately.

I'm not sure if Supporting Information is allowed, but if possible, consider presenting the additional analysis without Block 5 there. If there is no Supporting Information, maybe you find an elegant way of addressing it in the paper. If not, the current version is okay.

However, I'm confused about why information about mixed linear models was removed from the end of the Data Analyses section. The paper didn't claim that Anovas were run with LME4, but MLMs (still used in the "Did subjects make anticipatory steering adjustments?" section) were mentioned.

The main issue that still stands out is the small sample size. It's good that the focus on a small group of skilled individuals is clearly stated in the abstract, making it easy for any reader to understand.

Overall, I appreciate the improvements in this version, especially in addressing limitations and making the statistical analysis clearer.

Reviewer #3: 1.The abstract of this article needs to be summarized.

2.In conclusion，It is recommended that the authors point out the main future application scenarios of the main results of this paper.

7. PLOS authors have the option to publish the peer review history of their article (what does this mean?). If published, this will include your full peer review and any attached files.

Reviewer #1: **Yes: **Gianni Bremer

Reviewer #3: No

---

## [Author Response · Author response to Decision Letter 1]

28 Jan 2024

Response to reviewers

Manuscript number: PONE-D-23-23553R2

Manuscript title: Coordination of gaze and action during high-speed steering and obstacle avoidance

Authors: Powell, Marshall, Diaz, & Fajen 

We thank the reviewers for their careful reading of the revised manuscript. Our response to each comment can be found below in blue font. Changes to the manuscript are also indicated in the marked-up version. Note that line numbers below refer to those in the marked-up version of the manuscript with changes tracked. The line numbers in the unmarked version differ because MS Word includes deleted text in its count of line numbers.

Response to Editor instructions

Please review your reference list to ensure that it is complete and correct. If you have cited papers that have been retracted, please include the rationale for doing so in the manuscript text or remove these references and replace them with relevant current references. Any changes to the reference list should be mentioned in the rebuttal letter that accompanies your revised manuscript. If you need to cite a retracted article, indicate the article’s retracted status in the References list and also include a citation and full reference for the retraction notice.

We updated Reference 26, which was previously in press and is now published. We also added Reference 27 to support a point that was added in response to Reviewer 3’s second comment below.

Response to Reviewer #1 comments: 

The latest changes in the paper show a more careful consideration of limitations, a more transparent statistical analysis, and a discussion of "corner cutting behavior." The revised Anova is better explained, addressing assumptions and factors appropriately.

I'm not sure if Supporting Information is allowed, but if possible, consider presenting the additional analysis without Block 5 there. If there is no Supporting Information, maybe you find an elegant way of addressing it in the paper. If not, the current version is okay.

We added a paragraph toward the end of the section on analyses of basic performance measures (see Lines 395-401) that summarizes the analyses without Block 5. 

However, I'm confused about why information about mixed linear models was removed from the end of the Data Analyses section. The paper didn't claim that Anovas were run with LME4, but MLMs (still used in the "Did subjects make anticipatory steering adjustments?" section) were mentioned.

The reviewer is correct. The analyses in the section on anticipatory steering adjustments were run using the lmer function from the LME4 package. We added a sentence to the Data Analysis section to tell readers which R function was used (see Lines 325-327).

The main issue that still stands out is the small sample size. It's good that the focus on a small group of skilled individuals is clearly stated in the abstract, making it easy for any reader to understand. Overall, I appreciate the improvements in this version, especially in addressing limitations and making the statistical analysis clearer.

We are happy to hear that the reviewer is satisfied with the improvements and thank them for their helpful suggestions.

Response to Reviewer #3 comments: 

1.The abstract of this article needs to be summarized.

We are unsure what the reviewer means by this comment. The abstract is a summary of the article, so it is unclear to us what it would mean to summarize the abstract. We wondered whether the reviewer meant to write “shortened” rather than “summarized” but the abstract is already under the 300-word limit, so that wouldn’t make sense. We also wondered if they meant that the main take-away of the article needs to be summarized in the main text, but that wouldn’t make sense either since that is exactly the point of the last paragraph of the article. We apologize for not being able to respond to this comment, but we simply don’t understand what the reviewer is asking us to do.

2.In conclusion，It is recommended that the authors point out the main future application scenarios of the main results of this paper.

We explained how the findings could inform the development of models of driver performance under conditions in which visibility is degraded, which is an important but neglected area of driver behavior modeling (see Lines 679-683).

---

## [Editor Report · Decision Letter 2]

7 Feb 2024

Coordination of gaze and action during high-speed steering and obstacle avoidance

PONE-D-23-23553R2

Dear Dr. Brett,

We’re pleased to inform you that your manuscript has been judged scientifically suitable for publication and will be formally accepted for publication once it meets all outstanding technical requirements.

Kind regards,

Lei Zhang, PhD

Academic Editor

PLOS ONE

Additional Editor Comments (optional):

The revised manuscript can be accepted for publication.
---

## [Editor Report · Acceptance letter]

28 Feb 2024

PONE-D-23-23553R2 

PLOS ONE

Dear Dr. Fajen, 

I'm pleased to inform you that your manuscript has been deemed suitable for publication in PLOS ONE. Congratulations! Your manuscript is now being handed over to our production team.

Kind regards, 

on behalf of

Dr. Lei Zhang 

Academic Editor

PLOS ONE